# A CONNECTION BETWEEN ONE-STEP RL AND CRITIC REGULARIZATION IN REINFORCEMENT LEARNING

## ABSTRACT

Offline RL algorithms require careful regularization to avoid overfitting. One-step methods perform regularization by doing just a single step of policy improvement, while critic regularization methods do many steps of policy improvement with a regularized objective. In this paper, we draw a close connection between these methods: applying a multi-step critic regularization method with a regularization coefficient of 1 yields the same policy as one-step RL. While practical implementations violate our assumptions and critic regularization is typically applied with smaller regularization coefficients, our experiments nevertheless show that our analysis makes accurate, testable predictions about practical offline RL methods (CQL and one-step RL) with commonly-used hyperparameters.

## 1 INTRODUCTION

Reinforcement learning (RL) algorithms tend to perform better when regularized, especially when given access to only limited data, and especially in batch (i.e., offline) settings where the agent is unable to collect new experience. While RL algorithms can be regularized using the same tools as in supervised learning (e.g., weight decay, dropout), our focus will be on regularization methods unique to the RL setting (policy regularization, value regularization). Research on these sorts of regularization has grown significantly in recent years, yet theoretical work studying the tradeoffs between regularization methods remains limited (Vieillard et al., 2020).

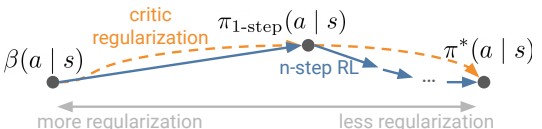

Figure 1: Both $n$-step RL and critic regularization can interpolate between behavioral cloning (left) and un-regularized RL (right) by varying the regularization parameter. Endpoints of these regularization paths are the same. We prove that these methods also obtain the same policy for an intermediate degree of regularization.

Many RL methods perform regularization, and can can be classified by whether they perform one or many steps of policy improvement. *One-step RL* methods (Brandfonbrener et al., 2021; Peng et al., 2019; Peters & Schaal, 2007; Peters et al., 2010) perform one step of policy iteration, updating the policy to choose actions the are best according to the Q-function of the behavioral policy. The policy is often regularized to not deviate far from the behavioral policy. In theory, policy iteration can take a large number of iterations ($\tilde{\mathcal{O}}(|\mathcal{S}||\mathcal{A}|/(1-\gamma))$ (Scherrer, 2013)) to converge, so one-step RL (one step of policy iteration) fails to find the optimal policy on most tasks. Empirically, policy iteration often converges in a smaller number of iterations (Sutton & Barto, 2018, Sec. 4.3), and the policy after just a single iteration can sometimes achieve performance comparable to multi-step RL methods (Brandfonbrener et al., 2021). *Critic regularization* methods modify the training of the value function such that it predicts smaller returns for unseen actions (Kumar et al., 2020; Chebotar et al., 2021; Yu et al., 2021; Hatch et al., 2022; Nachum et al., 2019; An et al., 2021; Bai et al., 2022; Buckman et al., 2020). See Appendix A for a discussion of prior methods.

These RL regularization methods appear quite distinct. Critic regularization typically involves solving a two-player game, whereby a policy predicts actions with high values while the critic decreases the values predicted for those actions. Prior work (Kumar et al., 2020) has argued that a benefit from this complexity is that the critic regularization ends up being propagated across time.

In this paper, we show that a certain type of actor and critic regularization can be equivalent, under some assumptions. The key idea is that, when using a certain TD loss, the regularized critic updates

converge not to the true Q-values, but rather the Q-values multiplied by an importance weight. For the critic, these importance weights mean that the Q-values end up estimating the expected returns of the behavioral policy ($Q^\beta$, as in many one-step methods (Peters et al., 2010; Peters & Schaal, 2007; Peng et al., 2019; Brandfonbrener et al., 2021)), rather than the expected returns of the optimal policy ($Q^\pi$). For the actor, these importance weights mean that the logarithm of the Q-values includes a term that looks like a KL divergence. So, optimizing the policy with these Q-values results in a standard form of actor regularization. Our analysis may help explain why prior work has found that one-step RL and critic regularization methods can perform similarly on some (Brandfonbrener et al., 2021; Emmons et al., 2021) (but not all (Kostrikov et al., 2021)) problems. While our results do not say whether users should regularize the actor or critic in practice, they hint that one-step RL methods may be a simpler way of achieving the theoretical and empirical properties of critic regularization on RL tasks that require strong regularization.

## 2 A CONNECTION BETWEEN ONE-STEP RL AND CRITIC REGULARIZATION

This section will introduce our main result. We first introduce a new actor critic method, not because we expect it to perform better than existing actor-critic methods, but rather because it allows us to make precise a connection between actor and critic regularization. Sec. 2.2 then states the main result. See Appendix B for a formal description of the notation and prior methods.

### 2.1 CLASSIFIER ACTOR CRITIC

The key to our analysis will be to treat Q-values like probabilities, so we define the critic loss in terms of a cross-entropy loss, similar to prior work (Kalashnikov et al., 2018; Eysenbach et al., 2021). Recalling that Q-values are positive (Sec. B.1), we transform the Q-values to have the correct range by using $\frac{Q}{Q+1} \in [0, 1)$. We will minimize the cross-entropy loss applied to the transformed Q-values:

$$\mathbb{E}_{p(s,a)} \left[ \mathcal{CE} \left( \frac{Q(s,a)}{Q(s,a)+1}; \frac{y^\pi(s,a)}{y^\pi(s,a)+1} \right) \right] \tag{1}$$

$$\overset{\text{const.}}{=} -\mathbb{E}_{p(s,a)} \left[ y^\pi(s,a) \log \frac{Q(s,a)}{Q(s,a)+1} + \log \frac{1}{Q(s,a)+1} \right] \triangleq \mathcal{L}_{\text{critic}}(Q, \pi), \tag{2}$$

In the last line we scale both the positive and negative term by $y^\pi(s,a) + 1$, a choice that does not change the optimal classifier but reduces notational clutter. In tabular settings, this new critic objective performs the same updates as Q-learning ($Q(s,a) \leftarrow r(s,a) + \gamma Q(s',a')$), so it is guaranteed to converge and produce the correct Q-values (see proof in Appendix Lemma 2.1). The actor objective is to maximize the expected *log* of the Q-values:

$$\max_\pi \mathcal{L}_{\text{actor}}(\pi) \triangleq \mathbb{E}_{p(s)\pi(a|s)} \left[ \log(Q^\pi(s,a)) \right] \quad \text{where } Q^\pi = \arg\min_Q \mathcal{L}_{\text{critic}}(Q, \pi). \tag{3}$$

While most actor-critic methods do not use the logarithm transformation, prior work on conditional behavioral cloning (e.g., (Savinov et al., 2018; Ding et al., 2019; Sun et al., 2019; Ghosh et al., 2020; Srivastava et al., 2019)) implicitly includes this transformation (Eysenbach et al., 2022). In the absence of additional regularization, the optimal policy $\pi(a \mid s) = \mathbb{1}(a = \arg\max_{a'} Q(s,a'))$ is the same as the optimal policy for the standard actor objective (without the logarithm). We will call this method *classifier actor critic*.

We next introduce a one-step version of this method, as well as a critic regularization variant that resembles CQL. While we will implicitly use a regularization coefficient of 1 below, Appendix E.1 discusses versions of classifier actor critic with varying degrees of regularization.

**One-step RL.** To make classifier actor critic resemble one-step RL (Brandfonbrener et al., 2021), we make two changes: estimating the value of the behavioral policy and adding a regularization term to the actor objective. To estimate the value of the behavioral policy, we modify the critic loss to sample the next action $a'$ from the behavioral policy (i.e., we use $y^\beta(s,a)$ rather than $y^\pi(s,a)$). We also regularize the policy by adding a relative entropy term to the actor loss, analogous to the reverse KL penalty used in one-step RL:

$$\max_\pi \mathbb{E}_{p(s)\pi(a|s)} \left[ \log Q^\beta(s,a) + \log \beta(a \mid s) - \log \pi(a \mid s) \right] \text{ where } Q^\beta(s,a) = \arg\min_Q \mathcal{L}_{\text{critic}}(Q, \beta). \tag{4}$$

In tabular settings, this critic objective estimates the Q-values for $\beta(a \mid s)$ (Appendix Lemma 2.1).

**Critic regularization.** To emulate CQL, we modify the critic loss (Eq. 2) by adding a penalty term that decreases the values for unseen actions. Whereas CQL applies this penalty to the Q-values directly, we will apply it to the logarithm of the Q-values:[1]

$$\mathcal{L}^r_{\text{critic}}(Q, \pi) \triangleq \mathcal{L}_{\text{critic}}(Q, \pi) + \lambda \bigg( \mathbb{E}_{p(s)\pi(a|s)} \left[ \log(Q(s,a)+1) \right] - \mathbb{E}_{p(s)\beta(a|s)} \left[ \log(Q(s,a)+1) \right] \bigg). \quad (5)$$

### 2.2 MAIN RESULT

To relate one-step RL to critic regularization, we start by analyzing the Q-values learned by both methods. We first show that the classifier critic converges to the correct Q-values:

**Lemma 2.1.** *In the tabular setting, applying the critic update to policy $\pi$ converges to $Q^\pi$:*

$$\arg\min_Q \mathcal{L}_{critic}(Q, \pi) = Q^\pi(s,a) \text{ for all states } s \text{ and actions } a. \quad (6)$$

Because one-step RL trains the critic using $\mathcal{L}_{\text{critic}}(Q, \beta)$, it learns Q-values corresponding to $Q^\beta(s,a)$.

When regularization is added to the critic updates, it learns different Q-values. Perhaps surprisingly, this regularization means that our estimates for the value of policy $\pi(a \mid s)$ look like the value of the original behavioral policy:

**Lemma 2.2.** *In the tabular setting, applying regularized critic updates with $\lambda = 1$ to policy $\pi$ converges to the Q-values for the behavioral policy ($\beta(a \mid s)$), weighted by the ratio of the behavioral and online policies:*

$$\arg\min_Q \mathcal{L}^r_{critic}(Q, \pi) = \frac{Q^\beta(s,a)\beta(a \mid s)}{\pi(a \mid s)} \text{ for all states } s \text{ and actions } a. \quad (7)$$

*Proof sketch.* The ratio $\frac{\beta(a|s)}{\pi(a|s)}$ above is an importance weight. Ordinarily, a TD backup for policy $\pi(a \mid s)$ would entail sampling an action $a \sim \pi(a \mid s)$. However, this importance weight means that TD backup is effectively performed by sampling an action $a \sim \beta(a \mid s)$. Such a TD backup resembles the TD backup for $\beta(a \mid s)$. The full proof is in Appendix D. □

Intuitively, this result says that critic regularization reweights the Q-values to assign higher values to in-distribution actions, where $\beta(a \mid s)$ is large. An unexpected part of this result is that the Q-values correspond to the behavioral policy. Said in other words, critic regularization added to a multi-step RL method (one using $y^\pi(s, a)$) yields the same critic as a one-step RL method (one using $y^\beta(s, a)$). Our main result is a direct corollary of this Lemma:

**Theorem 2.3.** *Let a behavioral policy $\beta(a \mid s)$ be given and let $Q^\beta(s, a)$ be the corresponding value function. Let $\pi(a \mid s)$ be an arbitrary policy (typically learned) with support constrained to $\beta(a \mid s)$ (i.e., $\pi(a \mid s) > 0 \implies \beta(a \mid s) > 0$). Let $Q^\pi_r(s, a)$ be the critic obtained by the regularized critic update (Eq. 5) to this policy with $\lambda = 1$. Then critic regularization results in the same policy as one-step RL:*

$$\mathbb{E}_{\pi(a|s)} \left[ \log Q^\pi_r(s,a) \right] = \mathbb{E}_{\pi(a|s)} \left[ \log Q^\beta(s,a) + \log \beta(a \mid s) - \log \pi(a \mid s) \right] \quad \text{for all states } s.$$

Since both forms of regularization result in the same objective for the actor, they must produce the same policy in the end. While prior work has mentioned that critic regularization implicitly regularizes the policy (Yu et al., 2021), this result shows that implicit regularization: under the assumptions stated above, the implicit regularization of critic regularization results in the exact same policy learning objective as one-step RL.

**Limitations.** Our theoretical analysis makes assumptions that may not always hold in practice. For example, our results use a critic loss based on the cross entropy loss, while most (but not all (Kalashnikov et al., 2018; Eysenbach et al., 2020b)) practical methods use the MSE. Our analysis assumes that critic regularization arrives at an equilibrium, and ignores errors introduced by function approximation and sampling.

---

[1]From a dimensional analysis perspective (Huntley, 1967), this choice makes sense because it allows the penalty term to have the same "units" as the critic loss: log Q-values. A second motivation for regularizing the logarithm is that the actor loss uses a logarithm.

**Extensions of the Analysis.** We extend this analysis in three ways. *First*, we also show that a similar connection can be established for lesser degrees of regularization ($\lambda < 1$) (see Appendix E.1). *Second*, we show that a similar connection holds for RL problems defined via success examples (Pinto & Gupta, 2016; Tung et al., 2018; Kalashnikov et al., 2021; Singh et al., 2019; Zolna et al., 2020; Calandra et al., 2017; Eysenbach et al., 2021) These results use existing actor-critic method, rather than classifier actor critic (see Appendix F). *Third*, we extend our analysis to multi-task settings by looking at goal-conditioned RL problems. We again show that a one-step version of a recent goal-conditioned RL method results in the same policy as a critic-regularized version of that same method (see Appendix G). Taken together, these extensions show that the connection between actor and critic regularization extends to other commonly-studied problem settings.

## 3 NUMERICAL SIMULATIONS

Our numerical simulations study whether the theoretical connection between actor and critic regularization holds empirically.

### 3.1 EXACT EQUIVALENCE WHEN USING CLASSIFIER ACTOR CRITIC

Our first experiment aims to validate our theoretical result under the required assumptions: when using classifier actor-critic as the RL algorithm, and when using a tabular environment. We use a $5 \times 5$ deterministic gridworld with 5 actions (up/down/left/right/nothing). In Fig. 3 we plot the action probabilities $\pi(a \mid s)$ for the policies produced by one-step RL and critic-regularized classifier actor-critic ($R^2 \geq 0.999$). This result confirms our theoretical results that these two methods should produce identical policies.

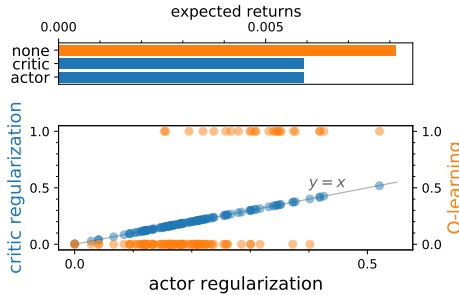

Figure 2: One-step RL and critic regularization produce identical policies.

### 3.2 PRACTICAL IMPLEMENTATIONS EXHIBIT SIMILAR BEHAVIOR

Based on our theoretical analysis, we predict that practical implementations of one-step RL and critic regularization will exhibit similar behavior, for a certain critic regularization coefficient. This section studies the tabular setting, and the following section will use a continuous control benchmark. For critic regularization, we used CQL (Kumar et al., 2020) together with soft value iteration; following (Brandfonbrener et al., 2021), we implement one-step RL (reverse KL) using Q-learning.

We designed a deterministic gridworld so one-step RL would fail to learn the optimal policy (see Fig. 3 *(left)*). If CQL interpolates between the behavioral policy (random) and the optimal policy, then the argmax action would always be the same as the action for $\pi^*$. Based on our analysis, we make a different prediction: that CQL will learn a policy similar to the one-step RL policy. We show results in Fig. 3 *(right)*, just showing the argmax action for visual clarity. The CQL policy takes actions away from both the high-reward state and the low reward state, similar to the behavioral policy but different from both the behavioral policy and the optimal policy. This experiment suggests that CQL can exhibit behavior similar to one-step RL. In Appendix C, we show additional experiments that show that one-step RL is most similar to CQL with a *moderate* regularization coefficient.

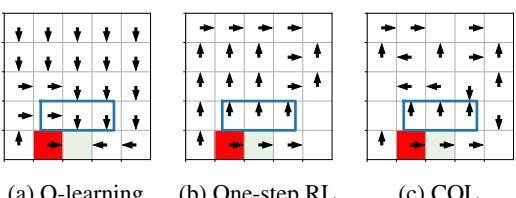

(a) Q-learning    (b) One-step RL    (c) CQL

Figure 3: **CQL can behave like one-step RL.** We design a gridworld so that one-step RL *(c)* learns a suboptimal policy. For the three cells highlighted in blue, the optimal policy *(b)* navigates towards the high-reward state (green) while the one-step RL policy *(c)* navigates away from the high-reward state. *(d)* CQL with a large regularization coefficient $\lambda = 10$ exhibits the same suboptimal behavior as one-step RL, taking actions that lead away from the high-reward states. For clarity, we only show the argmax action in each state; we omit the arrow when the argmax action is "do nothing".

### 3.3 Testing Predictions about Existing Offline RL Methods

Our final set of experiments studies whether our theoretical results can make accurate testable predictions about practically-used regularization methods in a setting where they are commonly used: offline RL benchmarks with continuous states and actions. For these experiments, we will use well-tuned implementations of CQL and one-step RL from Hoffman et al. (2020), using the default hyperparameters without modification. We made one change to the one-step RL implementation to makethe comparison more fair: because CQL learns two Q functions and takes the minimum (a trick introduced in Fujimoto et al. (2018)), we applied this same parametrization to the one-step RL implementation. Since offline RL methods can perform different on datasets of varying quality (Wang et al., 2020; Fujimoto & Gu, 2021; Paine et al., 2020; Wang et al., 2021; Fujimoto et al., 2019), we will repeat our experiments on four datasets from the D4RL benchmark (Fu et al., 2020).

**Lower bounds on Q-values.** One oft-cited benefit of critic regularization is that it has guarantees about value-estimation (Kumar et al., 2020): under appropriate assumptions, the learned value function will underestimate the discounted expected returns of the policy. Because our analysis shows a conenction between one-step RL and critic regularization, it raises the question of whether one-step RL methods have similar value-estimation properties. Taken

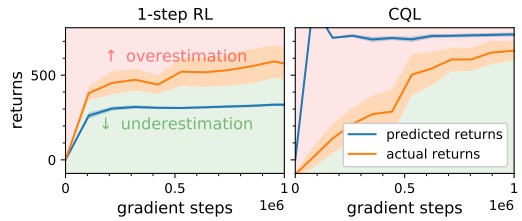

Figure 4: **Lower bound on Q-values.**

at face value, this hypothesis seems obvious: the behavioral critic estimates the value of the behavioral policy, so it should underestimate the value of any policy that is better than the behavioral policy. Despite this, the lower bound property of methods like one-step RL are rarely discussed, suggesting that it has yet to be widely appreciated. The results, shown in Fig. 4 for `medium-expert` dataset and Appendix Fig. 8 for all datasets, confirm our theoretical predictions while also questioning the claim that critic regularization methods are always preferable for ensuring underestimation.

**Critic regularization causes actor regularization.** Our analysis in Sec. 2 not only suggests that one-step RL methods might inherit properties of critic regularization (as studied in the previous section), but also suggests that critic regularization methods may behave like one-step methods. In particular, while critic regularization methods such as CQL do not explicitly regularize their actor, we hypothesize that they *implicitly* regularize the actor (Lemma 2.2), similar to how one-step RL methods explicitly regularize the actor. We measure the MSE between the action in the

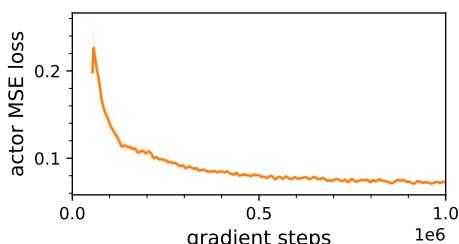

Figure 5: **CQL causes actor regularization.**

dataset and the action predicted by the learned policy. We show results on the `medium-expert` dataset in Fig. 5 and all datasets in Appendix Fig. 9. While directly regularizing the actor leads to MSE errors that are $\sim 3\times$ smaller, theses plot nevertheless show that critic regularization indirectly regularizes the actor.

## 4 Conclusion

In this paper, we drew a connection between two seemingly-distinct RL regularization methods: one-step RL and critic regularization. While our analysis made assumptions that are typically violated in practice, it nonetheless made accurate, testable predictions about practical methods with commonly-used hyperparameters: critic regularization methods can behave like one-step methods, and vice versa.

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

# A    RELATED WORK

Regularization has been applied to RL in many different ways (Neu et al., 2017; Geist et al., 2019), and features prominantly in offline RL methods (Lange et al., 2012; Levine et al., 2020). While RL algorithms can be regularized using the same techniques as in supervised learning (e.g., weight decay, dropout), our focus will be on regularization methods unique to the RL setting. Such RL-specific regularization methods can be categorized based on whether they regularize the actor or the critic.

One-step RL methods (Brandfonbrener et al., 2021; Gülçehre et al., 2020; Peters & Schaal, 2007; Peng et al., 2019; Peters et al., 2010; Wang et al., 2018) apply a single step of policy improvement to the behavioral policy. These methods first estimate the Q-values of the behavioral policy, either via regression or iterative Bellman updates. Then, these methods optimize the policy to maximize these Q-values minus an actor regularizer. Many goal-conditioned or task-conditioned imitation learning methods (Savinov et al., 2018; Ding et al., 2019; Sun et al., 2019; Ghosh et al., 2020; Paster et al., 2020; Yang et al., 2021; Srivastava et al., 2019; Kumar et al., 2019b; Chen et al., 2021; Lynch & Sermanet, 2021; Li et al., 2020; Eysenbach et al., 2020a) also fits into this mold (Eysenbach et al., 2022), yielding policies that maximize the Q-values of the behavioral policy while avoiding unseen actions. One-step methods are typically simple to implement and computationally efficient.

Critic regularization methods instead modify the objective for the Q-function so that it predicts lower returns for unseen actions (Kumar et al., 2020; Chebotar et al., 2021; Yu et al., 2021; Hatch et al., 2022; Nachum et al., 2019; An et al., 2021; Bai et al., 2022; Buckman et al., 2020). Critic regularization methods are typically more challenging to implement correctly and more computationally demanding (Kumar et al., 2020; Nachum et al., 2019; Bai et al., 2022; An et al., 2021), but can lead to better results on some challenging problems (Kostrikov et al., 2021) Our analysis will show that one-step RL is equivalent to a certain type of critic regularization. This result is surprising because it suggests that, for certain losses and hyperparameters, one-step RL yields the same solution as a multi-step RL method.

Some regularization methods do not fit exactly into these two categories. Methods like KL control regularize both the actor and the reward function (Geist et al., 2019; Ziebart, 2010; Haarnoja et al., 2018; Abdolmaleki et al., 2018; Wu et al., 2019; Jaques et al., 2019; Rezaeifar et al., 2022). Other methods methods only regularize the policy used in the critic updates (Fujimoto et al., 2019; Kumar et al., 2019a).

# B    BACKGROUND

We start by defining the single-task RL problem, and then introduce prototypical examples of one-step RL and critic regularization. We then introduce an actor critic algorithm we will use for our analysis.

## B.1    NOTATION

We assume an MDP with states $s$, actions $a$, initial state distribution $p_0(s_0)$, dynamics $p(s' \mid s, a)$, and reward function $r(s, a)$. Without loss of generality, we assume rewards are positive, adding a positive constant to all rewards can make them all positive without changing the optimal policy. We will learn a Markovian policy $\pi(a \mid s)$ to maximize the expected discounted sum of rewards:

$$\max_\pi \mathbb{E}_{\pi(\tau)} \left[ \sum_{t=0}^\infty \gamma^t r(s_t, a_t) \mid s_0 \sim p_0(s_0) \right],$$

where $\pi(\tau) = p(s_0) \prod_{t=0}^\infty \pi(a_t \mid s_t) p(s_{t+1} \mid s_t, a_t)$ is the probability of policy $\pi$ sampling an infinite-length trajectory $\tau = (s_0, a_0, \cdots)$. We define Q-values for policy $\pi(a \mid s)$ as

$$Q^\pi(s, a) = \mathbb{E}_{\pi(\tau)} \left[ \sum_{t=0}^\infty r(s_t, a_t) \mid s_0 = s, a_0 = a \right].$$

Note that the reward being positive implies that the Q-values are also positive, $Q^\pi(s, a) > 0$. Since we focus on the offline setting, we will consider two policies: $\beta(a \mid s)$ is the *behavioral* policy that collected the dataset, and $\pi(a \mid s)$ is the *online* policy output by the algorithm that attempts to maximize the rewards. We will use $p(s, a, s')$ to denote the empirical distribution of transitions in an offline dataset, and $p(s, a)$ and $p(s)$ denote the corresponding marginal distributions. The behavioral policy is defined as $\beta(a \mid s) = p(a \mid s)$.

## B.2    EXAMPLES OF REGULARIZATION IN RL

While actor and critic regularization methods can be implemented in many ways, we introduce two prototypical examples below to make our discussion more concrete.

**Example of one-step RL: Brandfonbrener et al. (2021).**    One-step RL first estimates the Q-values of the behavioral policy ($Q^\beta(s, a)$), and then optimizes the policy to maximize the Q-values minus a actor regularizer. While the actor regularizer can take different forms and the Q-values can be learned via regression, we will use a reverse KL regularizer and TD-style critic update so that the objective is similar to critic regularization:

$$\max_\pi \mathbb{E}_{p(s)\pi(a|s)} \left[ Q^\beta(s, a) + \lambda(\log \beta(a \mid s) - \log \pi(a \mid s)) \right] \tag{8}$$

$$\text{where} \quad Q^\beta = \arg\min_Q \mathbb{E}_{p(s,a)} \left[ \left( Q(s,a) - y^\beta(s,a) \right)^2 \right] \quad \text{and} \quad y^\beta(s,a) \triangleq r(s,a) + \gamma \mathbb{E}_{\substack{p(s'|s,a) \\ \beta(a'|s')}} \left[ Q(s',a') \right].$$

where $\lambda$ is the regularization coefficient and $\beta(a \mid s)$ is an estimate of the behavioral policy, typically learned via behavioral cloning. Here and in the rest of the paper, the TD targets $y(s, a)$ are not considered learnable (i.e., we would apply a stop-gradient operator). This one-step critic loss is different from the multi-step critic losses used in other RL methods (e.g., TD3, SVG(0)) because it uses the TD target $y^\beta(s, a)$ (corresponds to a fixed policy) rather than $y^\pi(s, a)$ (corresponding to a sequence of learned policies). One-step RL amounts to performing one step of policy iteration, rather than full policy optimization. While truncating the iterations of policy iteration can be suboptimal, it can also be interpreted as a form of early stopping regularization.

**Example of critic regularization: Kumar et al. (2020).**    CQL (Kumar et al., 2020) modifies the standard Bellman loss to include an additional term that decreases the values predicted for unseen actions. The actor objective is to maximize Q values; some CQL implementations also regularize the actor loss (Hoffman et al., 2020; Kumar et al., 2020)). The objectives can then be written as

$$\max_\pi \mathbb{E}_{p(s)\pi(a|s)} \left[ Q^\pi(s, a) \right] \tag{9}$$

$$\text{where} \quad Q = \arg\min_Q \mathbb{E}_{p(s,a)} \left[ (Q(s,a) - y^\pi(s,a))^2 \right] + \lambda \left( \mathbb{E}_{p(s)\pi(a|s)} \left[ Q(s,a) \right] - \mathbb{E}_{p(s)\beta(a|s)} \left[ Q(s,a) \right] \right).$$

The second term decreases the Q-values for unseen actions (those sampled from $\pi(a \mid s)$) while the third term increases the values predicted for seen actions (those sampled from the behavioral policy $\beta(a \mid s)$). Unlike standard temporal difference methods, the CQL updates resemble a competitive game between the actor and the critic. In practice, this cyclic dependency can create unstable learning (Kumar et al., 2020; Hoffman et al., 2020).

## B.3    HOW ARE THESE METHODS CONNECTED?

Prior work has observed that one-step methods and critic regularization methods perform similarly on many (Fujimoto & Gu, 2021; Emmons et al., 2021) (but not all (Kostrikov et al., 2021)) tasks. Despite the differences in objectives and implementations of these two methods (and, more broadly, the actor/critic regularization methods for which they are prototypes), are there deeper, unifying connections between the methods?

In the next section, we introduce a different actor-critic method that will allow us to draw a connection between one-step RL and critic regularization. We experimentally validate this equivalence in Sec. 3.1. Despite its difference from practically-used methods, such as one-step RL and CQL, we will show that it makes accurate predictions about the behavior of these practical methods (Sec. 3.2 and 3.3).

## C    ADDITIONAL EXPERIMENTS

**How often does one-step RL approximate CQL?**    To show that the results in Fig. 3 are not cherry-picked, we repeated this experiment using 100 MDPs that are structurally similar to that in Fig. 3, but where the locations of the high-reward and low reward state are randomized. In each randomly generated MDP, we determine whether CQL exhibits behavior similar to one-step RL by looking at the states where CQL takes actions that differ from the reward-maximizing actions (as

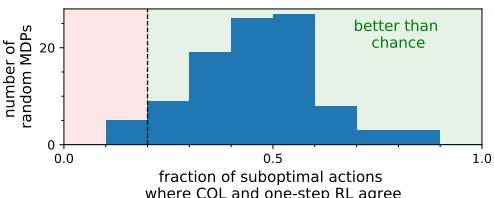

Figure 6: CQL and one-step RL take similar actions on most MDPs that resemble Fig. 3

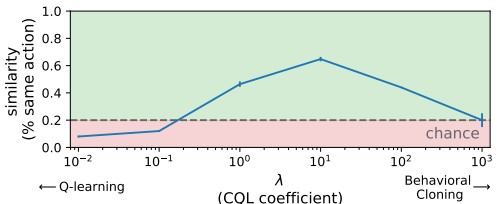

Figure 7: One-step RL is most similar to CQL with a moderate regularization coefficient.

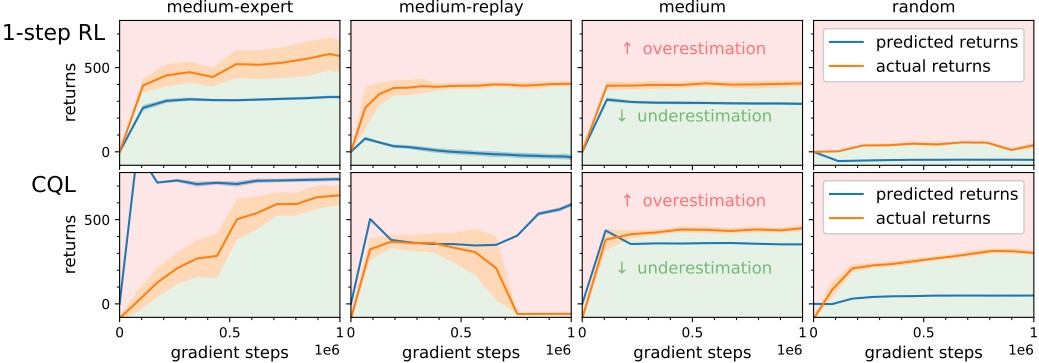

Figure 8: **Q-value under/over-estimation.** *(Top)* Experiments on benchmark datasets of varying quality show that one-step RL underestimates the Q-values. *(Bottom)* Despite the theoretical guarantees about critic regularization (CQL) yielding underestimates, in practice we observe that the values learned via critic regularization can sometimes overestimate the actual returns. We plot the mean and standard deviation across five random seeds.

determined by running Q-learning with unlimited data). Since there are five total actions, a random policy would have a similarity score of 20%. As shown in Fig. 6, the similarity score is significantly higher than chance for the vast majority of MDPs, showing that one-step RL and CQL($\lambda = 10$) produce similar policies on most such gridworlds.

**When does one-step RL approximate CQL?** Because one-step RL is highly regularized (policy iteration is truncated after just one step), one might imagine that it would be most similar to CQL with a very large regularization coefficient. To study this, we use the same environment (Fig. 3) and measure the fraction of states where one-step RL and CQL choose the same argmax action. As shown in Fig. 7, one-step RL is most similar to CQL with *moderate* regularization ($\lambda = 10$), and is less similar to CQL with a very strong regularization.

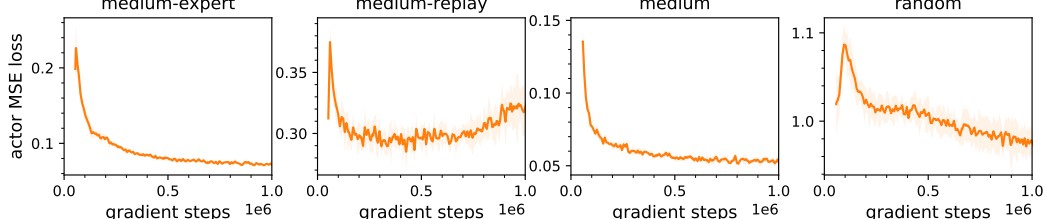

Figure 9: **Critic regularization causes actor regularization.** Performing critic regularization via CQL implicitly results in actor regularization, similar to one-step RL: the MSE between the predicted actions and the dataset actions decreases. We plot the mean and standard deviation across five random seeds.

## D  PROOFS

### D.1  PROOF OF LEMMA 2.1

*Proof.* As the cross entropy loss is minimized when the predictions equal the labels, updates for $\mathcal{L}_{\text{critic}}(Q, \pi)$ can be written as $\frac{Q(s,a)}{Q(s,a)+1} \leftarrow \frac{y^\pi(s,a)}{y^\pi(s,a)+1}$. If the updates are performed by averaging over all possible next states (e.g., in the tabular setting), these updates are equivalent to directly updating $Q(s,a) \leftarrow y^\pi(s,a) = r(s,a) + \gamma \mathbb{E}_{p(s'|s,a)\pi(a'|s')}[Q(s',a')]$, which is the standard policy evaluation update for policy $\pi(a \mid s)$. Thus, we can invoke the standard result that policy evaluation converges to $Q^\pi$ (Agarwal et al., 2019, Theorem 1.14.) to argue that updates for $\mathcal{L}_{\text{critic}}$ likewise converge to $Q^\pi$. $\qquad\square$

In this proof, the TD targets were the expectation over the next state and next action. If Eq. 2 were optimized using a single-sample estimate of this expectation, $\mathbf{y} = r(s,a) + \gamma Q(s',a')$, then the updates would be biased:

$$\frac{Q(s,a)}{Q(s,a)+1} \leftarrow \mathbb{E}\left[\frac{\mathbf{y}}{\mathbf{y}+1}\right] \leq \frac{\mathbb{E}[\mathbf{y}]}{\mathbb{E}[\mathbf{y}]+1} = \frac{y^\pi(s,a)}{y^\pi(s,a)+1}.$$

In settings with stochastic transitions or policies, these updates would result in estimating a lower bound on $Q^\pi(s,a)$.

### D.2  PROOF OF THEOREM 2.3

*Proof.* Our proof proceeds in three steps. First, we derive the update equations for the regularized critic update. That is, if we maintained a table of Q-values, what would the new value for $Q(s,a)$ be? Second, we show that these updates are equivalent to performing policy evaluation on a *reparametrized* critic $\tilde{Q}(s,a) = Q(s,a)\frac{\pi(a|s)}{\beta(a|s)}$. We invoke the standard results for policy evaluation to prove convergence that $\tilde{Q}(s,a)$ convergences. Finally, we undo the reparametrization to obtain convergence results for $Q(s,a)$.

**Step 0.** We start by rearranging the regularized critic objective:

$$\mathcal{L}^r_{\text{critic}}(Q,\pi) \triangleq \mathcal{L}_{\text{critic}}(Q,\pi) + \left(\mathbb{E}_{p(s)\pi(a|s)}[\log(Q(s,a)+1)] - \mathbb{E}_{p(s)\beta(a|s)}[\log(Q(s,a)+1)]\right)$$

$$= -\mathbb{E}_{p(s,a)}\left[y^\pi(s,a)\log\frac{Q(s,a)}{Q(s,a)+1} + \log\frac{1}{Q(s,a)+1}\right]$$

$$+ \left(\mathbb{E}_{p(s)\pi(a|s)}[\log(Q(s,a)+1)] - \mathbb{E}_{p(s)\beta(a|s)}[\log(Q(s,a)+1)]\right)$$

$$= -\mathbb{E}_{p(s,a)}\left[y^\pi(s,a)\log\frac{Q(s,a)}{Q(s,a)+1} + \log\frac{1}{Q(s,a)+1}\right]$$

$$- \left(\mathbb{E}_{p(s)\pi(a|s)}\left[\log\frac{1}{Q(s,a)+1}\right] + \mathbb{E}_{p(s)\beta(a|s)}\left[\log\frac{1}{Q(s,a)+1}\right]\right)$$

$$= -\mathbb{E}_{p(s,a)}\left[y^\pi(s,a)\log\frac{Q(s,a)}{Q(s,a)+1}\right] + \mathbb{E}_{p(s)\beta(a|s)}\left[\log\frac{1}{Q(s,a)+1}\right].$$

For the cancelation on the third line, we used the fact that $p(s,a) = p(s)\beta(a \mid s)$.

**Step 1.** To start, note that the regularized critic update is equivalent to a weighted classification loss: positive examples are sampled $(s,a) \sim p(s)\beta(a \mid s)$ and receive weight $\frac{y^\pi(s,a)}{y^\pi(s,a)+1}$, and negative examples are sampled $(s,a) \sim p(s)\pi(a \mid s)$ and receive weight $\frac{1}{y^\pi(s,a)+1}$. The Bayes' optimal classifier is given by

$$\frac{Q(s,a)}{Q(s,a)+1} = \frac{\frac{y^\pi(s,a)}{y^\pi(s,a)+1}p(s)\beta(a \mid s)}{\frac{y^\pi(s,a)}{y^\pi(s,a)+1}p(s)\beta(a \mid s) + \frac{1}{y^\pi(s,a)+1}p(s)\pi(a \mid s)} = \frac{y^\pi(s,a)\beta(a \mid s)}{y^\pi(s,a)\beta(a \mid s) + \pi(a \mid s)}.$$

Solving for $Q(s,a)$ on the left hand side, the optimal value for $Q(s,a)$ is given by

$$Q(s,a) = y^\pi(s,a)\frac{\beta(a \mid s)}{\pi(a \mid s)} = (r(s,a) + \mathbb{E}_{p(s'|s,a)\pi(a'|s')}[Q(s',a')])\frac{\beta(a \mid s)}{\pi(a \mid s)}. \tag{10}$$

This equation tells us what each update for the regularized critic loss does.

**Step 2.** To analyze these updates, we define $\tilde{Q}(s,a) \triangleq Q(s,a)\frac{\pi(a|s)}{\beta(a|s)}$. Then these updates can be written using $\tilde{Q}(s,a)$ as

$$\tilde{Q}(s,a)\frac{\beta(a \mid s)}{\pi(a \mid s)} = \left(r(s,a) + \mathbb{E}_{p(s'|s,a)\pi(a'|s')}\left[\tilde{Q}(s',a')\frac{\beta(a' \mid s')}{\pi(a' \mid s')}\right]\right)\frac{\beta(a \mid s)}{\pi(a \mid s)}, \quad (11)$$

which can be simplified to

$$\tilde{Q}(s,a) = r(s,a) + \mathbb{E}_{p(s'|s,a)\beta(a'|s')}\left[\tilde{Q}(s',a')\right]. \quad (12)$$

Note that the ratio $\frac{\beta(a'|s')}{\pi(a'|s')}$ inside the expectation acts like an importance weight, so that the expectation over $\pi(a' \mid s')$ becomes an expectation over $\beta(a' \mid s')$. Thus, the regularized critic updates are equivalent to perform policy evaluation on $\tilde{Q}(s,a)$. An immediately consequence is that the regularized critic updates converge, and they converge to $\tilde{Q}^*(s,a) = Q^\beta(s,a)$.

**Step 3.** Finally, we translate these convergence results for $\tilde{Q}(s,a)$ into convergence results for $Q(s,a)$. Written in terms of the original Q-values, we see that the optimal critic for the regularized critic update is

$$Q^*(s,a) = \tilde{Q}^*(s,a)\frac{\beta(a \mid s)}{\pi(a \mid s)} = Q^\beta(s,a)\frac{\beta(a \mid s)}{\pi(a \mid s)}. \quad (13)$$

$\square$

# E   VARYING THE REGULARIZATION COEFFICIENT

While our main analysis (Theorem 2.3)showed that regularization and critic regularization yield the same policy when these regularizers are applied with a certain strength, in practice the strength of regularization is controlled by a hyperparameter. This hyperparameter raises a question: *does the connection between one-step RL and critic regularization hold for different values of this hyperparameter?*

In this section, we show that there remains a precise connection between actor and critic regularization, even for different values of this hyperparameter. This result not only suggests that the connection is stronger than initially suggested by the main result. Proving this connection also helps highlight how many regularization methods can be cast from a similar mold.

## E.1   A REGULARIZATION COEFFICIENT.

We start by modifying the actor regularizer and critic regularizer introduced in Sec. 2.1 to include an additional hyperparameter.

**Mixture policy.**   Both the actor and critic losses will make use of a mixture policy, $(1 - \lambda)\pi(a \mid s) + \lambda\beta(a \mid s)$, where $\lambda \in [0,1]$ will be a hyperparameter. Larger values of $\lambda$ yield a mixture policy that is closer to the behavioral policy; this will correspond to higher degrees of regularization. Mixtures of policies are commonly used in practice (Kumar et al., 2020, Appendix F),(Villaflor et al., 2020, Eq. 11), (Finn et al., 2016, Sec. 4.3) (Lyu et al., 2022) (Hazan et al., 2019, Eq. 2.5), even though it rarely appears in theoretical offline RL literature. Indeed, because critic regularization resembles a two-player zero-sum game, mixture policies might even be *required* to find a (Nash) equilibrium of the critic regularizer (Nash, 1951).

**$\lambda$-weighted critic loss.**   With this concept of a mixture policy, we define the $\lambda$-weighted actor and critic regularizers. For the $\lambda$-weighted critic loss, we will change how the TD targets are computed. Instead of sampling the next action from $\pi$ or $\beta$, we will sample the next action from a $\lambda_{\text{TD}}$-weighted combination of these two policies, reminiscent of how prior work has regularized the actions sampled for the TD backup (Fujimoto et al., 2019; Zhou et al., 2020):

$$y^{\lambda_{\text{TD}}} \triangleq y^{(1-\lambda)\pi+\lambda\beta}(s,a) = r(s,a) + \gamma\mathbb{E}_{\substack{p(s'|s,a)\\(1-\lambda_{\text{TD}})\pi(a|s)+\lambda_{\text{TD}}\beta(a|s)}}[Q(s',a')].$$

When introducing one-step RL in Sec. 2.1, we used $\lambda_{\text{TD}} = 1$.

Using this TD target, the $\lambda$-weighted critic loss can now be written as a combination of the un-regularized objective (Eq. 2) plus the regularized objective (Eq. 5):

$$\mathcal{L}_{\text{critic}}^r(Q, \lambda_{\text{critic}}) \triangleq (1 - \lambda_{\text{critic}}) \left( -\mathbb{E}_{p(s,a)} \left[ \frac{y^{\lambda_{\text{TD}}}(s,a)}{y^{\lambda_{\text{TD}}}(s,a) + 1} \log \frac{Q(s,a)}{Q(s,a) + 1} + \frac{1}{y^{\lambda_{\text{TD}}}(s,a) + 1} \log \frac{1}{Q(s,a) + 1} \right] \right)$$

$$+ \lambda \left( -\mathbb{E}_{\substack{p(s,a) \\ a^- \sim \pi(\cdot | s)}} \left[ \frac{y^{\lambda_{\text{TD}}}(s,a)}{y^{\lambda_{\text{TD}}}(s,a) + 1} \log \frac{Q(s,a)}{Q(s,a) + 1} + \frac{1}{y^{\lambda_{\text{TD}}}(s,a) + 1} \log \frac{1}{Q(s,a) + 1} \right] \right)$$

$$= -\mathbb{E}_{\substack{p(s,a) \\ a^- \sim (1 - \lambda_{\text{critic}})\pi(\cdot | s) + \lambda_{\text{critic}}\beta(\cdot | s)}} \left[ \frac{y^{\lambda_{\text{TD}}}(s,a)}{y^{\lambda_{\text{TD}}}(s,a) + 1} \log \frac{Q(s,a)}{Q(s,a) + 1} + \frac{1}{y^{\lambda_{\text{TD}}}(s,a) + 1} \log \frac{1}{Q(s,a^-) + 1} \right].$$

$$(14)$$

The second line rewrites this objective: the first term looks the same as the original "positive" term in the critic objective, while the "negative" term uses actions sampled from a mixture of the current policy and the behavioral policy. When $\lambda_{\text{critic}} = 1$, we recover the regularized critic loss introduced in Sec. 2.1.

**$\lambda$-weighted actor loss.** Finally, the strength of the actor regularizer can be controlled by changing the reverse KL penalty. While it may seem like changing the reward scale would varying the strength of the actor loss, this is not the case for classifier actor critic because of the $\log(\cdot)$ in the actor loss. Instead, we will relax the reverse KL penalty between the learned policy $\pi(a \mid s)$ and the behavioral policy $\beta(a \mid s)$ so that only the mixture policy only needs to be close to behavioral policy:

$$\mathcal{L}_{\text{actor}}^r(\pi, \lambda_{\text{KL}}) \triangleq \mathbb{E}_{p(s)\pi(a|s)} \left[ \log Q(s,a) + \log \beta(a \mid s) - \log \left( (1 - \lambda_{\text{KL}})\pi(a \mid s) + \lambda_{\text{KL}}\beta(a \mid s) \right) \right]. \quad (15)$$

As indicated on the second line, replacing $\beta(a \mid s)$ with the mixture policy has an effect similar to that of decreasing the weight applied to the KL penalty. The approximation on the second line is determined by the Jensen Gap (Abramovich & Persson, 2016; Gao et al., 2017). When introducing one-step RL in Sec. 2.1, we used $\lambda_{\text{KL}} = 1$, together with $\lambda_{\text{TD}} = 1$.

In summary, the strength of the actor and critic regularizers can be controlled through additional hyperparameters ($\lambda_{\text{critic}}, \lambda_{\text{TD}}, \lambda_{\text{KL}}$). Indeed, it is typical for offline RL methods to require many hyperparameters (Brandfonbrener et al., 2021; Lu et al., 2021; Paine et al., 2020; Wu et al., 2019), and performance is sensitive to their settings. However, the close connection that we have shown between actor and critic regularizers allows us to decrease the number of hyperparameters.

### E.2 ANALYSIS

In our main result (Thm. 2.3), we showed that one-stel RL and critic regularization are equivalent when $\lambda_{\text{critic}} = \lambda_{\text{TD}} = \lambda_{\text{KL}} = 1$. This is a large value for the regularization strength, and we now consider what happens for smaller degrees of regularization: is there still a connection between one-step RL and critic regularization?

The following theorem will prove that this is the case. In particular, applying critic regularization with coefficient $\lambda_{\text{critic}}$ yields the same policy as applying one-step RL with $\lambda_{\text{TD}} = \lambda_{\text{KL}} = \lambda_{\text{critic}}$. That is, there is a very simple recipe for converting the hyperparameters for critic regularization into the hyperparameters for one-step RL.

**Theorem E.1.** *Let policy $\pi(a \mid s)$ be given, let $Q^\beta(s,a)$ be the Q-function of the behavioral policy, and let $Q_r^{\lambda_{TD}}(s,a,\lambda_{critic})$ be the critic obtained by the $\lambda_{critic}$-weighted regularized critic update (Eq. 14) using TD targets $y^{\lambda_{TD}}(s,a)$. If $\lambda_{critic} = \lambda_{TD} = \lambda_{KL}$, then the $\lambda_{KL}$-weighted actor loss (Eq. 15) is equivalent to the un-regularized policy objective using the regularized critic:*

$$\mathbb{E}_{p(s)\pi(a|s)} \left[ \log Q(s,a) + \log \beta(a \mid s) - \log \left( (1 - \lambda_{KL})\pi(a \mid s) + \lambda_{KL}\beta(a \mid s) \right) \right]$$

$$= \mathbb{E}_{\pi(a|s)} \left[ \log Q_r^{\lambda_{TD}}(s,a,\lambda_{critic}) \right] \qquad \textit{for all states } s.$$

While we used the cross entropy loss for this result, it turns out that the result also holds for the more standard MSE loss (we omit the proof for brevity).

**Limitations.** Before presenting the proof in Sec. E.3, we discuss a few limitations of this result. Like the rest of the analysis in this paper, the form of the critic regularizer is different from that often used in practice. Additionally, our analysis assumes ignores many sources of errors (e.g., sampling, function approximation), and assumes that each objective is optimized exactly.

### E.3 Proof of Theorem E.1

*Proof.* We start by defining the fixed point of the $\lambda$-weighted regularized critic loss. Like in the single-task setting, this loss resembles a weighted classification problem, so we can write down the Bayes' optimal classifier as

$$\frac{Q(s,a)}{Q(s,a)+1} = \frac{\frac{y^{\lambda_{\mathrm{TD}}(s,a)}}{y^{\lambda_{\mathrm{TD}}(s,a)}+1}p(s)\beta(a \mid s)}{\frac{y^{\lambda_{\mathrm{TD}}(s,a)}}{y^{\lambda_{\mathrm{TD}}(s,a)}+1}p(s)\beta(a \mid s) + \frac{1}{y^{\lambda_{\mathrm{TD}}(s,a)}+1}p(s)((1-\lambda_{\mathrm{critic}})\pi(a \mid s) + \lambda_{\mathrm{critic}}\beta(a \mid s))}$$

$$= \frac{y^{\lambda_{\mathrm{TD}}}(s,a)\beta(a \mid s)}{y^{\lambda_{\mathrm{TD}}}(s,a)\beta(a \mid s) + (1-\lambda_{\mathrm{critic}})\pi(a \mid s) + \lambda_{\mathrm{critic}}\beta(a \mid s)}.$$

Solving for $Q(s,a)$ on the left hand side, the optimal value for $Q(s,a)$ is given by

$$Q(s,a) = y^{\lambda_{\mathrm{TD}}}(s,a)\frac{\beta(a \mid s)}{(1-\lambda_{\mathrm{critic}})\pi(a \mid s) + \lambda_{\mathrm{critic}}\beta(a \mid s)}$$

$$= (r(s,a) + \mathbb{E}_{p(s'\mid s,a),a'\sim(1-\lambda_{\mathrm{TD}}),\pi(\cdot\mid s')+\lambda_{\mathrm{TD}}\beta(\cdot\mid s)}[Q(s',a')])\frac{\beta(a \mid s)}{(1-\lambda_{\mathrm{critic}})\pi(a \mid s) + \lambda_{\mathrm{critic}}\beta(a \mid s)}. \tag{16}$$

Note that the next action $a'$ is sampled from a mixture policy defined by $\lambda_{\mathrm{TD}}$. This equation tells us what each update for the $\lambda$-weighted regularized critic loss does.

To analyze these updates, we define

$$\tilde{Q}(s,a) \triangleq Q(s,a)\frac{(1-\lambda_{\mathrm{critic}})\pi(a \mid s) + \lambda_{\mathrm{critic}}\beta(a \mid s)}{\beta(a \mid s)}.$$

Like before, the ratio $\frac{\beta(a'\mid s')}{(1-\lambda_{\mathrm{TD}})\pi(a'\mid s')+\lambda_{\mathrm{TD}}\beta(a'\mid s')}$ can act like an importance weight. When $\lambda_{\mathrm{TD}} = \lambda_{\mathrm{critic}}$, then this importance weight cancels with the sampling distribution, providing the following identity:

$$\mathbb{E}_{p(s'\mid s,a),a'\sim(1-\lambda_{\mathrm{TD}}),\pi(\cdot\mid s')+\lambda_{\mathrm{TD}}\beta(\cdot\mid s)}[Q(s',a')]$$

$$= \mathbb{E}_{p(s'\mid s,a),a'\sim(1-\lambda_{\mathrm{TD}}),\pi(\cdot\mid s')+\lambda_{\mathrm{TD}}\beta(\cdot\mid s)}\left[\tilde{Q}(s,a)\frac{\beta(a \mid s)}{(1-\lambda_{\mathrm{critic}})\pi(a \mid s) + \lambda_{\mathrm{critic}}\beta(a \mid s)}\right]$$

$$= \mathbb{E}_{p(s'\mid s,a),a'\sim\beta(\cdot\mid s')}[\tilde{Q}(s,a)].$$

Substituting this identity in Eq. 16, we can write the updates using $\tilde{Q}(s,a)$:

$$\tilde{Q}(s,a)\frac{\beta(a \mid s)}{(1-\lambda_{\mathrm{critic}})\pi(a \mid s) + \lambda_{\mathrm{critic}}\beta(a \mid s)}$$

$$= \left(r(s,a) + \mathbb{E}_{p(s'\mid s,a),a'\sim\beta(\cdot\mid s')}[\tilde{Q}(s,a)]\right)\frac{\beta(a \mid s)}{(1-\lambda_{\mathrm{critic}})\pi(a \mid s) + \lambda_{\mathrm{critic}}\beta(a \mid s)},$$

which can be simplified to

$$\tilde{Q}(s,a) = r(s,a) + \mathbb{E}_{p(s'\mid s,a),a'\sim\beta(\cdot\mid s')}[\tilde{Q}(s,a)].$$

We then translate these convergence results for $\tilde{Q}(s,a)$ into convergence results for $Q(s,a)$. Written in terms of the original Q-values, we see that the optimal critic for the regularized critic update is

$$Q^*(s,a) = Q^\beta(s,a)\frac{\beta(a \mid s)}{(1-\lambda_{\mathrm{critic}})\pi(a \mid s) + \lambda_{\mathrm{critic}}\beta(a \mid s)}. \tag{17}$$

Note that this holds for any value of $\lambda_{\mathrm{critic}} = \lambda_{\mathrm{TD}} \in [0,1]$. This result suggests that two common forms of regularization, decreasing the values predicted at unseen actions and regularizing the actions

used in the TD backup, can produce the same effect: a critic that estimates the Q-values of the behavioral policy (multiplied by some importance weight).

Finally, substitute this Q-function into the un-regularized actor loss, we see that the result is equivalent to the $\lambda$-weighted actor loss:

$$\mathbb{E}_{p(s)\pi(a|s)}\left[\log Q^*(s,a)\right] = \mathbb{E}_{p(s)\pi(a|s)}\left[\log Q^\beta(s,a) + \underbrace{\log \beta(a\mid s) - \log\left((1-\lambda_{\mathrm{KL}})\pi(a\mid s) + \lambda_{\mathrm{KL}}\beta(a\mid s)\right)}_{\lambda\text{-weighted actor regularizer}}\right]$$

$\square$

## F  REGULARIZATION FOR GOAL-CONDITIONED PROBLEMS

Like single-task RL problems, goal-conditioned RL problems have also been approached with both one-step methods (Ghosh et al., 2020; Ding et al., 2019; Sun et al., 2019) and critic regularization (Chebotar et al., 2021). In these problems, the aim is to learn a goal-conditioned policy $\pi(a \mid s, s_g)$ that maximizes the expected discounted sum of goal-conditioned rewards $r_g(s,a)$, where goals are sampled $s_g \sim p_g(s_g)$:

$$\max_\pi \mathbb{E}_{p_g(s_g)}\mathbb{E}_{\pi(\tau|s_g)}\left[\sum_{t=0}^{\infty}\gamma^t r_g(s_t, a_t)\right].$$

We will use the goal-conditioned reward function $r_g(s,a) = p(s' = s_g \mid s, a)$, which is defined in terms of the environment dynamics. In settings with discrete states, maximizing this reward function is equivalent to maximizing the sparse indicator reward function ($r_g(s,a) = \mathbb{1}(s_g = s)$).

In this section, we show that one-step RL and critic regularization are equivalent for a certain goal-conditioned actor-critic method. Unlike our analysis in the single-task setting, this analysis here uses an existing method, C-learning (Eysenbach et al., 2020b). C-learning is a TD method that already makes use of the cross entropy loss for training the critic:

$$\max_Q (1-\gamma)\mathbb{E}_{p(s,a,s')}\left[\log\frac{Q(s,a,s_g=s')}{Q(s,a,s_g=s')+1}\right] + \gamma\mathbb{E}_{p(s,a)p_g(s_g)}\left[y^\pi(s,a,s_)\log\frac{Q(s,a,s_g)}{Q(s,a,s_g)+1}\right]$$
$$+ \mathbb{E}_{p(s,a)p_g(s_g)}\left[\log\frac{1}{Q(s,a,s_g=s')+1}\right],$$

where $y^\pi(s,a,s_g) = \mathbb{E}_{p(s'|s,a)\pi(a'|s',s_g)}\left[Q(s',a',s_g)\right]$ serves the role of the TD target.

The first two terms increase the Q-values while the last term decreases the Q-values. The actor is updated to maximize the Q-values. While this objective for the actor can be written in many ways, we will write it as maximizing a log ratio because it will allow us to draw a precise equivalence between actor and critic regularization:

$$\max_\pi \mathbb{E}_{p_g(s_g)p(s)\pi(a|s,s_g)}\left[\log Q(s,a,s_g)\right]$$

We will now consider variants of C-learning that incorporate actor and critic regularization.

**One-step RL.**  We will consider a variant of C-learning that resembles one-step RL (Brandfonbrener et al., 2021). The critic update will be similar to before, but the next-actions sampled for the TD updates will be sampled from the *marginal* behavioral policy:

$$\max_Q (1-\gamma)\mathbb{E}_{p(s,a,s')}\left[\log\frac{Q(s,a,s_g=s')}{Q(s,a,s_g=s')+1}\right] + \gamma\mathbb{E}_{p(s,a)p_g(s_g)}\left[y^\beta(s,a,s_)\log\frac{Q(s,a,s_g)}{Q(s,a,s_g)+1}\right]$$
$$+ \mathbb{E}_{p(s,a)p_g(s_g)}\left[\log\frac{1}{Q(s,a,s_g=s')+1}\right],$$

where $y^\beta(s,a,s_g) = \mathbb{E}_{p(s'|s,a)\beta(a'|s')}[Q(s',a',s_g)]$. The actor update will be modified to include a reverse KL divergence:

$$\max_\pi \mathbb{E}_{p(s)p_g(s_g)\pi(a|s,s_g)}\left[\log Q(s,a,s_g) + \log\beta(a\mid s) - \pi(a\mid s, s_g)\right]. \tag{18}$$

Note that we are regularizing the policy to be similar to the average behavioral policy, $\beta(a \mid s)$. Compared to regularization towards a goal-conditioned behavioral policy $\beta(a \mid s, s_g)$, this choice gives the policy additional flexibility: when trying to reach goal $s_g$, it is allowed to take actions that were not taken by $\beta(a \mid s, s_g)$, as long as they were taken by the behavioral policy when trying to reach some other goal $s'_g$.

**Critic regularization.** To regularize the critic, we will modify the "negative" term in the C-learning objective to use actions sampled from the policy:

$$\max_Q (1 - \gamma) \mathbb{E}_{p(s,a,s')} \left[ \log \frac{Q(s, a, s_g = s')}{Q(s, a, s_g = s') + 1} \right] \tag{19}$$

$$+ \gamma \mathbb{E}_{p(s,a)p_g(s_g)} \left[ y^\pi(s, a, s_g) \log \frac{Q(s, a, s_g)}{Q(s, a, s_g) + 1} \right] \tag{20}$$

$$+ \mathbb{E}_{p(s)p_g(s_g)a \sim \pi(\cdot|s,s_g)} \left[ \log \frac{1}{Q(s, a, s_g) + 1} \right]. \tag{21}$$

### F.1 ANALYSIS FOR GOAL-CONDITIONED PROBLEMS

Like in the single-task setting, these two forms of regularization yield the same fixed points:

**Theorem F.1.** *Let policy $\pi(a \mid s, s_g)$ be given, let $Q^\beta(s, a, s_g)$ be the Q-values for the marginal behavioral policy $\beta(a \mid s)$ and let $Q_r^\pi(s, a, s_g)$ be the critic obtained by the regularized critic update (Eq. 21). Then performing regularized policy updates (Eq. 18) using the behavioral critic is equivalent to the un-regularized policy objective using the regularized critic:*

$$\mathbb{E}_{\pi(a|s,s_g)} \left[ \log Q^\beta(s, a, s_g) + \log \beta(a \mid s) - \log \pi(a \mid s, s_g) \right] = \mathbb{E}_{\pi(a|s,s_g)} \left[ \log Q_r^\pi(s, a, s_g) \right]$$

*for all states $s$ and goals $s_g$.*

*Proof.* We start by determining the fixed point of critic-regularized C-learning. Like in the single-task setting, the C-learning objective resembles a weighted-classification problem, so we can write down the Bayes' optimal classifier as

$$\frac{Q(s, a, s_g)}{Q(s, a, s_g) + 1} = \frac{((1 - \gamma)p(s' = s_g \mid s, a) + \gamma p(s = s_g)y(s', s_g))\beta(a \mid s)}{((1 - \gamma)p(s' = s_g \mid s, a) + \gamma p(s = s_g)y(s', s_g))\beta(a \mid s) + p(s_g)\pi(a \mid s, s_g)}.$$

Solving for $Q(s, a, s_g)$ on the left hand side, the optimal value for $Q(s, a, s_g)$ is given by

$$Q(s, a, s_g) = ((1 - \gamma)p(s' = s_g \mid s, a) + \gamma p(s = s_g)y(s', s_g)) \frac{\beta(a \mid s)}{\pi(a \mid s, s_g)}$$

This tells us what each critic-regularized C-learning update does.

To analyze these updates, we define $\tilde{Q}(s, a, s_g) \triangleq Q(s, a, s_g) \frac{\pi(a|s,s_g)}{\beta(a|s)}$. Then these updates can be written using $\tilde{Q}(s, a, s_g)$ as

$$\tilde{Q}(s, a, s_g) \frac{\beta(a \mid s)}{\pi(a \mid s, s_g)} = \left( (1 - \gamma)p(s' = s_g \mid s, a) + \gamma \mathbb{E}_{p(s'|s,a)\pi(a'|s',s_g)} \left[ \tilde{Q}(s', a', s_g) \frac{\beta(a' \mid s')}{\pi(a' \mid s', s_g)} \right] \right) \frac{\beta(a \mid s)}{\pi(a \mid s, s_g)}.$$

These updates can be simplified to

$$\tilde{Q}(s, a, s_g) = (1 - \gamma)p(s' = s_g \mid s, a) + \gamma \mathbb{E}_{p(s'|s,a)\beta(a'|s')} \left[ \tilde{Q}(s', a', s_g) \right].$$

Like before, the ratio $\frac{\beta(a'|s')}{\pi(a'|s',s_g)}$ inside the expectation acts like an importance weight. Thus, the regularized critic updates are equivalent to perform policy evaluation on $\tilde{Q}(s, a, s_g)$. Note that this is estimating the probability that the *average* behavioral policy $\beta(a \mid s)$ reaches goal $s_g$; this is *not* the probability that a goal-directed behavioral policy $\beta(a \mid s, s_g)$ reaches the goal.

Finally, we translate these convergence results for $\tilde{Q}(s, a, s_g)$ into convergence results for $Q(s, a, s_g)$. Written in terms of the original Q-values, we see that the optimal critic for the regularized critic update is

$$Q^*(s, a, s_g) = \tilde{Q}^*(s, a, s_g) \frac{\beta(a \mid s)}{\pi(a \mid s, s_g)} = Q^{\beta(\cdot|\cdot)}(s, a, s_g) \frac{\beta(a \mid s)}{\pi(a \mid s, s_g)}.$$

Thus, critic regularization implicitly regularizes the actor objective so that it is the same objective as one-step RL:

$$\mathbb{E}_{p(s),s_g \sim p(s),\pi(a|s,s_g)} \left[\log Q^*(s,a,s_g)\right]$$
$$= \mathbb{E}_{p(s),s_g \sim p(s),\pi(a|s,s_g)} \left[\log Q^{\beta(\cdot|\cdot)}(s,a,s_g) + \log \beta(a \mid s) - \log \pi(a \mid s, s_g)\right].$$

$\square$

## G  REGULARIZATION FOR EXAMPLE-BASED CONTROL PROBLEMS

While specifying tasks in terms of reward functions is standard for MDPs, it can be difficult for real-world applications of RL. So, prior work has looked at specifying tasks by goal states (as in the previous section) or sets of states representing good outcomes (Pinto & Gupta, 2016; Tung et al., 2018; Fu et al., 2018). In addition to requiring more flexible and user-friend forms of task specification, these algorithms targeted at real-world applications often demand regularization. In the same way that prior goal-conditioned RL algorithms have employed critic regularization, so too have prior example-based control algorithms (Singh et al., 2019; Hatch et al., 2022). In this section, we extend our analysis to regularization of an example-based control algorithm. Again, we will show that a certain form of critic regularization is equivalent to regularizing the actor.

We first define the problem of example-based control (Fu et al., 2018). In these problems, the agent is given a small collection of states $s \sim p_e(s)$, which are examples of successful outcomes. The aim is to learn a policy $\pi(a \mid s)$ that maximizes the probability of reaching a success state:

$$\max_\pi \mathbb{E}_{p(s_g)} \mathbb{E}_{\pi(\tau|s_g)} \left[\sum_{t=0}^\infty \gamma^t p_e(s_t)\right].$$

Note that this objective function is exactly equivalent to a reward-maximization problem, with a reward function $r(s,a) = p_e(s_t)$.

In this section, we show that one-step RL and critic regularization are equivalent for a certain example-based control algorithm. Unlike our analysis in the single-task setting, this analysis here uses an existing method, RCE (Eysenbach et al., 2021). RCE is a TD method that already makes use of the cross entropy loss for training the critic:

$$\max_Q (1-\gamma)\mathbb{E}_{p_e(s)\beta(a|s)} \left[\log \frac{Q(s,a)}{Q(s,a)+1}\right] + \mathbb{E}_{p(s,a)} \left[\gamma y^\pi(s,a) \log \frac{Q(s,a)}{Q(s,a)+1} + \log \frac{1}{Q(s,a)+1}\right],$$

where $y^\pi(s,a) = \mathbb{E}_{p(s'|s,a)\pi(a'|s')}[Q(s',a')]$ serves the role of the TD target. The first two terms increase the Q-values while the last term decreases the Q-values. The actor is updated to maximize the Q-values. While this objective for the actor can be written in many ways, we will write it as maximizing a log ratio because it will allow us to draw a precise equivalence between actor and critic regularization:

$$\max_\pi \mathbb{E}_{p(s)\pi(a|s)} \left[\log Q(s,a)\right]$$

We will now consider variants of RCE that incorporate actor and critic regularization.

**One-step RL.**  We will consider a variant of RCE that resembles one-step RL (Brandfonbrener et al., 2021). The critic update will be similar to before, but the next-actions sampled for the TD updates will be sampled from the behavioral policy:

$$\max_Q (1-\gamma)\mathbb{E}_{p_e(s)\beta(a|s)} \left[\log \frac{Q(s,a)}{Q(s,a)+1}\right] + \mathbb{E}_{p(s,a)} \left[\gamma y^\beta(s,a) \log \frac{Q(s,a)}{Q(s,a)+1} + \log \frac{1}{Q(s,a)+1}\right],$$

where $y^\beta(s,a) = \mathbb{E}_{p(s'|s,a)\beta(a'|s')}[Q(s',a')]$. The actor update will be modified to include a reverse KL divergence:

$$\max_\pi \mathbb{E}_{p(s),\pi(a|s)} \left[\log Q(s,a) + \log \beta(a \mid s) - \pi(a \mid s)\right]. \tag{22}$$

**Critic regularization.**  To regularize the critic, we will modify the "negative" term in the RCE objective to use actions sampled from the policy:

$$(1-\gamma)\mathbb{E}_{p_e(s)\beta(a|s)} \left[\log \frac{Q(s,a)}{Q(s,a)+1}\right] + \mathbb{E}_{p(s,a),a^- \sim \pi(\cdot|s)} \left[\gamma y^\pi(s,a) \log \frac{Q(s,a)}{Q(s,a)+1} + \log \frac{1}{Q(s,a^-)+1}\right], \tag{23}$$

### G.1 ANALYSIS FOR EXAMPLE-BASED CONTROL PROBLEMS

Like in the single-task setting, these two forms of regularization yield the same fixed points:

**Theorem G.1.** *Let policy $\pi(a \mid s)$ be given, let $Q^\beta(s, a)$ be the Q-values for the behavioral policy $\beta(a \mid s)$ and let $Q_r^\pi(s, a)$ be the critic obtained by the regularized critic update (Eq. 23). Then performing regularized policy updates (Eq. 22) using the behavioral critic is equivalent to the un-regularized policy objective using the regularized critic:*

$$\mathbb{E}_{\pi(a|s)} \left[ \log Q^\beta(s, a) + \log \beta(a \mid s) - \log \pi(a \mid s) \right] = \mathbb{E}_{\pi(a|s)} \left[ \log Q_r^\pi(s, a) \right]$$

*for all states $s$.*

*Proof.* We start by determining the fixed point of critic-regularized RCE. Like in the single-task setting, The RCE objective resembles a weighted-classification problem, so we can write down the Bayes' optimal classifier as

$$\frac{Q(s, a)}{Q(s, a) + 1} = \frac{((1 - \gamma)p_e(s) + \gamma y^\pi(s, a))\beta(a \mid s)}{((1 - \gamma)p_e(s) + \gamma y^\pi(s, a))\beta(a \mid s) + \pi(a \mid s)}.$$

Solving for $Q(s, a)$ on the left hand side, the optimal value for $Q(s, a)$ is given by

$$Q(s, a) = ((1 - \gamma)p_e(s) + \gamma y^\pi(s, a)) \frac{\beta(a \mid s)}{\pi(a \mid s)}$$

This tells us what each critic-regularized RCE update does.

To analyze these updates, we define $\tilde{Q}(s, a) \triangleq Q(s, a) \frac{\pi(a|s)}{\beta(a|s)}$. Then these updates can be written using $\tilde{Q}(s, a)$ as

$$\tilde{Q}(s, a) \frac{\beta(a \mid s)}{\pi(a \mid s)} = \left( (1 - \gamma)p_e(s) + \gamma \mathbb{E}_{p(s'|s,a)\pi(a'|s')} \left[ \tilde{Q}(s', a') \frac{\beta(a' \mid s')}{\pi(a' \mid s')} \right] \right) \frac{\beta(a \mid s)}{\pi(a \mid s)}.$$

These updates can be simplified to

$$\tilde{Q}(s, a) = (1 - \gamma)p_e(s) + \gamma \mathbb{E}_{p(s'|s,a)\beta(a'|s')} \left[ \tilde{Q}(s', a') \right].$$

Like before, the ratio $\frac{\beta(a'|s')}{\pi(a'|s')}$ inside the expectation acts like an importance weight. Thus, the regularized critic updates are equivalent to perform policy evaluation on $\tilde{Q}(s, a)$.

Finally, we translate these convergence results for $\tilde{Q}(s, a)$ into convergence results for $Q(s, a)$. Written in terms of the original Q-values, we see that the optimal critic for the regularized critic update is

$$Q^*(s, a) = \tilde{Q}^*(s, a) \frac{\beta(a \mid s)}{\pi(a \mid s)} = Q^\beta(s, a) \frac{\beta(a \mid s)}{\pi(a \mid s)}.$$

Thus, critic regularization implicitly regularizes the actor objective so that it is the same objective as one-step RL:

$$\mathbb{E}_{p(s),\pi(a|s)} \left[ \log Q^*(s, a) \right] = \mathbb{E}_{p(s),\pi(a|s)} \left[ \log Q^\beta(s, a) + \log \beta(a \mid s) - \log \pi(a \mid s) \right].$$

$\square$

## H EXPERIMENTAL DETAILS

### H.1 TABULAR EXPERIMENTS

**Implementing critic regularization for classifier actor critic.** The objective for critic regularization in contrastive actor critic (Eq. 5) is nontrivial to optimize because of the cyclic dependency between the policy and the critic: simply alternating between optimizing the actor and the critic does not converge. In our experiments, we update the critic using an exponential moving average of the policy, as proposed in Wen et al. (2021). We found that this decision was sufficient for ensuring convergence. When applying CQL in the tabular setting (Figures 3 and 6), we did not do this because soft value iteration represents the policy implicitly in terms of the value function.

**Fig. 2** *(left)*    The initial state and goal state are located in opposite corners. The reward function is +1 for reaching the goal and 0 otherwise. We use a dataset of 20 trajectories, 50 steps each, collected by a random policy. We use $\gamma = 0.95$ and train for 20k full-batch updates, using a learning rate of 1e-2. The Q table is randomly initialized using a standard normal distribution.

**Fig. 2** *(center)*    The initial state and goal state are located in adjacent corners. The goal state has a reward of +3.5, the states between the initial state and goal state have a reward +1, and all other states (including the initial state) have a reward of +2. We use a dataset of 20 trajectories, 50 steps each, collected by a random policy. We use $\gamma = 0.95$ and train for 20k full-batch updates, using a learning rate of 1e-2. The Q table is randomly initialized using a standard normal distribution.

**Fig. 2** *(right)*    The initial state and goal state are located in adjacent corners. The reward is +0.01 at the goal state and 0 otherwise. We use a dataset of 1 trajectories with 10 steps, which traces the following path:

$$[(0,0), (1,0), (1,1), (1,2), (1,3), (1,4), (0,4), (0,4), (0,4), (0,4)].$$

We use $\gamma = 0.95$ and train for 10k full-batch updates, using a learning rate of 1e-2. The Q table is randomly initialized using a standard normal distribution.

**Fig. 3**    There is a bad state (reward of $-10$) next to the optimal state (reward of $+1$), so the behavioral policy navigates away from the optimal state. We generate 10 trajectories of length 100 from a uniform random policy. We use $\gamma = 0.95$ and train each method for 10k full-batch updates. The Q table is randomly initialized using a standard normal distribution. One-step RL performs SARSA updates while CQL performs soft value iteration (as suggested in the CQL paper).

**Fig. 6**    We generate 100 random variants of Fig. 3 by randomly sampling the high-reward state and low-reward state (without replacement). The datasets are generated in the same way.

**Fig. 7**    We use the same environment and dataset as in Fig. 3, but train the CQL agent with varying values of $\lambda$, each with 5 random seeds. We train the one-step RL agent for 5 random seeds. For each point on the X axis of Fig. 7, we compare compute $5 \times 5$ pairwise comparisons and report the mean and standard deviation.

## H.2    Continuous control experiments

For the experiments in Figures 8 and 9, we used the implementation of one-step RL (reverse KL) and CQL provided by Hoffman et al. (2020). We choose this implementation because it is well tuned and uses similar hyperparameters for the two methods. As mentioned in the main text, the only change we made to the implementation was adding the twin-Q trick to one-step RL, such that it matched the critic architecture used by CQL. We did not change any of the other hyperparameters, including hyperparameters controlling the regularization strength.

