# OpenReview forum: "A Connection between One-Step Regularization and Critic Regularization in Reinforcement Learning"
_NeurIPS.cc/2022/Workshop/Offline_RL — Offline RL Workshop NeurIPS 2022_

### Official Review · Reviewer_FpNJ · 2022-10-18
**Paper review**

**Rating:** 7
**Confidence:** 3

**Review:**

### Summary

This paper establishes an equivalence between certain forms of actor and critic regularization in the batch setting, where data is collected from a behavioral policy. Essentially, if you learn a critic (Q function) via critic regularization (which discounts rewards for actions that are unlikely under the behavioral policy), and you use this critic to learn a policy, then you end up learning the same policy you would learn with actor-regularized 1-step RL (which regularizes the policy toward the behavioral policy). The main repercussion of this finding is that 1-step RL may be a more efficient way to find the same policy you would otherwise get from critic regularization.

### Strengths

* In terms of grammar/syntax and flow, the paper is very well written. (More on clarity in Weaknesses.)
* The authors do a pretty good job of explaining what each result means.
* The paper is upfront about its limitations -- namely, the strength of its assumptions and applicability of the results.
* Connections are made to existing algorithms, so the setup/assumptions used seem less contrived.
* To my knowledge, this work is novel and not obvious. Though, to be completely honest, I am not well versed in the (vast) literature on actor-critic methods.

### Weaknesses

* The motivation for the central question, and why the result is significant, could be elaborated on. If I've understood correctly, the takeaway is that one could do 1-step RL instead of CQL, and that this is somehow more efficient. I think the paper says this (in so many words), but it could be made a bit clearer, or emphasized more. To better convey the significance, the paper could, e.g., quantify the computation/time saved (either analytically or empirically) by performing 1-step RL instead of critic regularization.
* While the paper is well written from a technical perspective (that is, one doesn't need to struggle through typos/bad grammar to decypher each sentence), I found myself struggling to understand the _concepts_ being conveyed. I think this is partly a result of editing what is clearly a longer paper down to just 5 pages; and partly due to my lack of background in actor-critic methods. (Also, I must admit, I did not read the appendices, which may have provided key details or background info.)
* The Gridworld experiment doesn't seem very convincing, as the example world seems a bit arbitrary and the result is presented in a way that requires extrapolation from the learned policies. This would be more convincing with a more quantitative presentation.
* The experiments in Sec 3.3 had too little detail and discussion for me to take much away from them. (I realize this is an unfortunate consequence of the page limit.)

### Other comments
* The paper analyzes a _classifier actor-critic_ framework in which, crucially, the Q-values and rewards are normalized to the interval $[0, 1]$ via the transformation $Q / (Q + 1)$. This allows for the critic loss to be defined as a cross-entropy. However, the paper makes a simplification to the cross-entropy by multiplying it by $y^\pi(s,a) + 1$, arguing that minimizing this objective would result in the same model. I'm having a hard time understanding why this is true. $Q^\pi$ is trained to minimize the _expected value_ of the loss over draws of $s, a \sim p(s, a)$, so $y^\pi(s,a)$ is not a constant that can be simply pulled outside the argmin. Maybe there's some implicit monotonicity argument? I wish the paper would explain this in more depth, as I got stuck on this point.
* I caught a couple typos in Sec 3.3: "makethe" -> "make the"; "conenction" -> "connection".

---

### Official Review · Reviewer_rp6b · 2022-10-20

**Rating:** 6
**Confidence:** 3

**Review:**

Recent work showed that one-step Reinforcement Learning (RL) methods can achieve similar empirical performance to multi-step regularized methods in offline RL problems. This paper studies this phenomenon theoretically and shows, under some assumptions, that both classes of methods can lead to the same learned policy and value functions.

I think that the topic investigated in the paper is significant and interesting for the offline community. I have a few issues with how the paper is written, as I found that many assumptions were not well justified. For instance, in the paper, it is not explained why we need to adopt a cross-entropy loss for the analysis or why the actor is maximizing the logarithm of Q values. On the latter, the authors cite work on Upside-Down RL, which however never uses a critic, so I am not sure if the choice is well justified. Also, the choice of applying a logarithm in eq. 5 seems arbitrary. Do the theoretical results hold without such an assumption?

Without proper justification, I find it difficult to understand if the theoretical results are significant or if they are just a trivial consequence of the many assumptions made by the authors.

The experiments' goal is to show that an implementation of CQL behaves similarly to one-step RL. I am confused by Figure 3, which should show how CQL has the same behavior as one-step RL. The authors show that the actions in 3 states are the same across both algorithms, however, the actions in all other states are quite different. If the two algorithms induce the same policy, why is there such a big difference?

Figure 4 also shows very different values in the value function prediction, while the authors claim that the result confirms their theoretical analysis.

I recommend the authors properly justify the technical assumptions made in the paper and clarify the experimental results.